# No Long-Term Superiority of Cord-Derived PRP over Autologous PRP in Knee Osteoarthritis: A Prospective Cohort Study

**DOI:** 10.3390/jfmk10020138

**Published:** 2025-04-21

**Authors:** Michele Coviello, Antonella Abate, Giuseppe Maccagnano, Alessandro Geronimo, Elio Caiaffa, Vittorio Nappi, Vincenzo Caiaffa, Giuseppe Solarino

**Affiliations:** 1Department of Clinical and Experimental Medicine, Orthopaedics Unit, Faculty of Medicine and Surgery, University of Foggia, Policlinico Riuniti di Foggia, 71122 Foggia, Italy; 2Orthopaedic and Traumatology Unit, “Di Venere” Hospital, 70131 Bari, Italy; 3Department of Basic Medical Sciences, Orthopaedic and Trauma Unit, Neurscience and Sense Organs, School of Medicine, AOU Consorziale Policlinico, University of Bari “Aldo Moro”, 70124 Bari, Italy

**Keywords:** knee osteoarthritis, platelet-rich plasma, autologous PRP, umbilical cord PRP, regenerative medicine, intra-articular injection, pain relief, functional outcomes

## Abstract

**Background:** Knee osteoarthritis (OA) is a progressive joint disorder characterized by pain, stiffness, and functional impairment. Platelet-rich plasma (PRP) has been widely studied as a biological treatment for OA, with autologous PRP (A-PRP) being the most commonly used formulation. Recently, umbilical cord-derived PRP (C-PRP) has emerged as a potential alternative due to its hypothesized higher regenerative potential. However, evidence supporting its superiority over A-PRP remains limited. This study aims to compare the efficacy and safety of C-PRP and A-PRP in terms of pain relief and functional improvement over a 12-month follow-up period. **Methods:** This prospective cohort study included 84 patients with mild-to-moderate knee OA (Kellgren–Lawrence grades I–III), into two groups: 44 patients received a single intra-articular injection of C-PRP, and 40 received A-PRP. Pain and functional outcomes were assessed at baseline, 3, 6, 9, and 12 months using the Visual Analog Scale (VAS), Western Ontario and McMaster Universities Osteoarthritis Index (WOMAC), and Knee Injury and Osteoarthritis Outcome Score (KOOS). Statistical analysis was performed using the Mann–Whitney U, Exact Fisher test, repeated measures general linear model (GLM) and multivariate logistic regression. **Results:** Both C-PRP and A-PRP led to significant pain reduction and functional improvement over 12 months (*p* < 0.01 for both groups). Short-term analysis (3–6 months) showed slightly greater pain relief in the C-PRP group (VAS, *p* = 0.03 at 3 months), but this difference diminished at later time points. By 9 and 12 months, no significant differences were observed between the two groups in any clinical outcome measures (VAS, WOMAC, KOOS; *p* > 0.05). No serious adverse events were reported, and both treatments were well tolerated. **Conclusions:** This study found no long-term superiority of C-PRP over A-PRP in terms of pain relief or functional improvement in knee OA. While C-PRP showed a transient advantage in early pain relief, both treatments demonstrated similar clinical outcomes at 12 months. Given the limited scientific evidence supporting C-PRP and its higher logistical costs, A-PRP should remain the preferred PRP therapy for knee OA. Further randomized controlled trials with longer follow-up periods are needed to confirm these findings.

## 1. Introduction

Knee osteoarthritis (OA) is a degenerative joint disorder characterized by progressive cartilage degradation, synovial inflammation, and subchondral bone remodeling, leading to pain, stiffness, and functional limitations [1,2]. It is a significant cause of disability, particularly in older adults, with a rising prevalence due to aging populations and increasing obesity rates. Current management strategies primarily focus on symptom relief through physical therapy, nonsteroidal anti-inflammatory drugs (NSAIDs), intra-articular corticosteroids, and viscosupplementation. However, these approaches do not address the underlying degenerative process, often resulting in the need for surgical intervention in advanced cases [3,4].

In recent years, biologic intra-articular injections, particularly platelet-rich plasma (PRP), have gained attention as potential disease-modifying therapies. PRP is an autologous blood-derived product containing platelets, growth factors, and cytokines involved in tissue repair and inflammation modulation [5]. Several studies have suggested that PRP injections may provide pain relief and functional improvement in knee OA, with effects comparable to or better than corticosteroids and hyaluronic acid in certain patient populations [6,7]. However, A-PRP may also contain pro-inflammatory cytokines [8], and its quality is influenced by various patient-related factors, such as platelet count and function, concurrent pharmacological treatments [9], and patient age—often advanced in individuals affected by osteoarthritis (OA) [10,11]. To address these limitations, platelet-rich plasma derived from umbilical cord blood (C-PRP) has recently emerged as a promising alternative. Its use is supported by its allogeneic origin, which makes it independent of the biological variability of the recipient. C-PRP offers the theoretical advantage of containing higher concentrations of growth factors [12], and from a qualitative standpoint, it has been shown to exhibit increased levels of anti-inflammatory molecules compared to A-PRP, which instead is richer in pro-inflammatory factors [13].

Notably, C-PRP contains NK group 2 molecules, which suppress Natural Killer (NK), Natural Killer T (NKT), and T cell activity—a mechanism believed to contribute to maternal–fetal immune tolerance. It also presents elevated levels of various cytokines and growth factors, particularly Interleukin-10 (IL-10), which has been associated with reductions in pain intensity as measured by VAS scores [14]. Due to these properties, C-PRP has already found successful application across several medical specialties [15], including ophthalmology—for the treatment of corneal disorders and severe dry eye—and dermatology, where it has been used to promote the healing of chronic wounds and skin lesions. Nonetheless, the body of evidence supporting its use in orthopedic conditions, such as hip osteoarthritis, remains limited, despite increasing interest in the therapeutic potential of homologous PRP sources [16].

Another relevant focus of research pertains to the duration of the analgesic effect, as well as the potential to induce meaningful anatomical modifications in the progression of the degenerative osteoarthritic process. Some preliminary findings suggest that C-PRP may provide short-term improvements in pain and function, but its long-term efficacy and safety are not well established [8]. Recent studies demonstrate that PRP not only effectively reduces symptoms over a prolonged period, for at least 24 months, but also yields substantial benefits in terms of significantly attenuating tibiofemoral cartilage loss. Notably, MRI follow-ups conducted over a span of 5 years consistently reveal marked reductions in tibiofemoral cartilage degeneration following PRP treatment, underscoring its potential as a durable therapeutic intervention for KOA [14,17].

Despite its theoretical advantages, the scientific evidence supporting C-PRP remains limited. Only a few studies have investigated its clinical effectiveness, and long-term comparative data with A-PRP are lacking [18,19]. Furthermore, factors such as cryopreservation, donor variability, and potential immunogenic responses require further investigation before C-PRP can be considered a standard treatment option [8,9]. A-PRP and C-PRP represent two promising therapeutic options for the treatment of symptomatic knee and hip osteoarthritis. However, both have limitations and variables primarily related to preparation methods as well as intrinsic properties; the type of patient and the severity of osteoarthritis further influence the outcomes, particularly the duration of the pain-relieving effect.

This study aims to address this gap by comparing the clinical efficacy and safety of single intra-articular injections of C-PRP and A-PRP in patients with mild-to-moderate knee OA, with a 12-month follow-up to assess both short-term and long-term outcomes.

## 2. Materials and Methods

### 2.1. Patients and Study Design

This prospective cohort study was conducted with approval from the local Ethics Committee (delib. 0104, approved on 12 November 2019) and focused on patients with low-to-moderate knee osteoarthritis (OA). The study was carried out in collaboration with the Transfusion Medicine Unit at Di Venere Regional Hospital and the Puglia Cord Blood Bank. Cord blood (CB) units were collected from mothers who provided informed consent, and processing was completed within 48 h of collection. Standardized protocols for platelet concentration and centrifugation were followed. The resulting cord blood platelet concentrate (CBPC) units were transferred into storage bags and cryopreserved at temperatures below −40 °C without the use of cryoprotectants. For clinical application, the CB platelet gel (CBPG) was thawed in a 37 °C water bath and activated using a standardized technique [10,11,20].

The study population included individuals aged 37–79 years, recruited between December 2019 and June 2022. All participants provided written informed consent prior to treatment. Inclusion criteria consisted of symptomatic knee OA with daily pain persisting for at least three months and unresponsive to pain medication, absence of meniscal rupture confirmed by MRI, and a Kellgren–Lawrence (K-L) radiographic classification of grade I–III. Radiographic staging was independently assessed by a senior radiologist [21] using standard knee X-rays in standing anteroposterior and horizontal lateral projections, supplemented by MRI findings. Exclusion criteria included the presence of condylar or tibial plateau bone marrow edema on MRI, significant axial deviation (valgus > 10° or varus > 5°) of the affected limb, recent intra-articular hyaluronic acid or steroid injections within the preceding six months, ipsilateral hip or ankle arthritis, and a history of malignancy.

Blood type and ABO-Rh compatibility were determined for all participants. Patients were randomized into two groups using a predefined algorithm (http://www.randomizer.org, accessed on 1 December 2019). Following randomization, the treatment options, including their potential benefits and risks, were explained to the patients. Those who disagreed with their assigned group were permitted to switch therapies, prior to study initiation. A total of 84 patients were enrolled: 44 received a single intra-articular injection of C-PRP (5 mL), and 40 received a single intra-articular injection of A-PRP (5 mL). All injections were administered by two experienced orthopedic surgeons, each with over five years of expertise in knee intra-articular procedures. The injections were performed using a 5 mL syringe via an anterolateral approach at the medial joint line with the knee flexed at 90°. Post-procedure, patients were advised to avoid physical activity for 72 h, with acetaminophen (up to 3 g/day) permitted for pain management.

Prior to enrollment, patients underwent a comprehensive clinical evaluation, including a complete blood count and screening for transmissible infections such as HIV, HBV, and HCV. The study adhered to the Strengthening the Reporting of Observational Studies in Epidemiology (STROBE) guidelines, and Figure 1 provides a detailed flowchart of patient recruitment, enrollment, and evaluation [22].

### 2.2. Methods

Cord blood (CB) units obtained from the Puglia Cord Blood Bank were included in the study if they met the following criteria: total nucleated cell counts ≥ 1.5 × 10^9^, platelet counts ≥ 150 × 10^9^/L, and a volume ≥ 50 mL. Within 48 h of collection, the CB units underwent an initial soft centrifugation process. The resulting platelet-rich plasma was then subjected to high-speed centrifugation to produce cord blood platelet concentrate (CBPC) with a target platelet concentration of 800–1200 × 10^9^/L. The CBPC was cryopreserved without cryoprotectants at temperatures below −40 °C. The estimated preparation cost for each CBPC unit was €160.92, approximately 20% higher than the cost of autologous PRP (A-PRP). A single technician required approximately two hours to prepare four CBPC units [23]. For the preparation of autologous PRP (A-PRP), the Arthrex Angel System (Arthrex^®^, Naples, FL, USA) was used. The process began with the collection of 150 mL of autologous blood into a sterile bag. The blood was centrifuged at 1800 rpm for 15 min, after which the plasma and buffy coat were transferred to a second bag via a closed circuit, effectively removing red blood cells. The second bag was then centrifuged at 3500 rpm for 10 min, and the supernatant was discarded to yield approximately 20 mL of PRP. The goal was to achieve a 4–5-fold increase in total platelet count compared to baseline, resulting in a mean platelet concentration of 1000 × 10^9^/L ± 20%. The PRP was divided into three aliquots of 5 mL each, with an additional sample reserved for testing. The aliquots were stored at −30 °C. Prior to each treatment, the PRP was thawed in a dry thermostat at 37 °C for 300 s and then transferred to the outpatient clinic in a thermal bag to prevent light exposure. Immediately before injection, 10% calcium gluconate was added to activate the platelets. An independent orthopedic surgeon blinded to the PRP type injected performed clinical evaluations. The Visual Analog Scale (VAS) [24], the Western Ontario and McMaster Universities Arthritis Index (WOMAC) [25], and the Knee Injury and Osteoarthritis Outcome Score (KOOS) [26] were collected. These assessments were conducted at baseline (T0) and at follow-ups of 3 months (T1), 6 months (T2), 9 months (T3), and 12 months (T4). Clinical evaluations were conducted by an independent orthopedic surgeon who was blinded to the type of PRP administered. The following outcome measures were collected at baseline (T0) and at follow-up intervals of 3 months (T1), 6 months (T2), 9 months (T3), and 12 months (T4): Visual Analog Scale (VAS) [24]: A subjective tool for assessing acute and chronic pain. Patients marked a point on a 10-cm line ranging from “no pain” to “worst pain” to indicate their pain level. Western Ontario and McMaster Universities Osteoarthritis Index (WOMAC) [25]: A standardized questionnaire used to evaluate joint stiffness, pain, and physical function in patients with hip or knee osteoarthritis. Knee Injury and Osteoarthritis Outcome Score (KOOS) [26]: A self-administered questionnaire assessing five domains: pain, symptoms, activities of daily living, sport and recreation function, and knee-related quality of life. The primary endpoint of the study was the efficacy of the treatments, as measured by changes in the clinical outcome scales (VAS, WOMAC, and KOOS). The secondary endpoint was the safety of C-PRP and A-PRP treatments, which was evaluated by monitoring the incidence of treatment-related adverse events.

### 2.3. Statistical Analysis

This research was conducted as a prospective clinical study. Data collection and analysis were performed using SPSS software (version 23; IBM^®^ Inc., Armonk, NY, USA). A post hoc power analysis was conducted using the Visual Analog Scale (VAS) score as the primary outcome measure. Based on a standard deviation of 0.6 points, an α level of 0.05, and a power of 80%, it was determined that 27 participants per group would be required to detect statistically significant differences [2].

Descriptive statistics were calculated for the entire sample, including knee osteoarthritis severity, follow-up data, and baseline characteristics. Categorical variables were expressed as numbers or percentages, while continuous variables were reported as means and standard deviations. The Shapiro–Wilk test was used to assess the normality of the data. Since the data were not normally distributed, non-parametric tests were employed for analysis. The Mann–Whitney U test was used to compare outcomes between groups at the same time points, while the Fisher’s exact Chi-square test was applied to evaluate associations between dichotomous variables.

To assess changes over time, a repeated measures general linear model (GLM) with Sidak correction for multiple comparisons was utilized. Additionally, a multivariate logistic regression analysis was performed to identify variables significantly associated with a 3-point reduction in the VAS score at the final follow-up in both groups. The best-fit model was selected using a backward likelihood ratio test, with a removal threshold set at *p* > 0.10 and an entry threshold at *p* < 0.05. The Hosmer and Lemeshow goodness-of-fit test was used to evaluate the appropriateness of the logistic regression model.

Statistical significance was defined as a *p*-value of less than 0.05.

## 3. Results

The study included a total of 84 patients, with 44 receiving C-PRP and 40 receiving A-PRP. Of these, 20 (45.5%) in the C-PRP group and 22 (55%) in the A-PRP group were female. The left knee was affected in 19 (43.2%) patients in the C-PRP group and 15 (37.5%) patients in the A-PRP group. Baseline characteristics, including demographic and clinical variables, were comparable between the two groups, with no significant differences observed (Table 1).

The Mann–Whitney U test revealed significant differences between the C-PRP and A-PRP groups, though these differences were primarily observed during the short-term follow-up period. Specifically, the C-PRP group demonstrated greater improvements compared to the A-PRP group at 3 and 6 months. Statistically significant differences were noted in Visual Analog Scale (VAS) scores at 3 months (*p* < 0.05), as well as in Western Ontario and McMaster Universities Osteoarthritis Index (WOMAC) and Knee Injury and Osteoarthritis Outcome Score (KOOS) scores at both 3 and 6 months. However, by the final 12-month assessment, no significant differences were observed between the two groups, as outlined in Table 2.

Significant improvements were observed in both the C-PRP and A-PRP groups from baseline (T0) to the final follow-up (*p* = 0.01 for both groups), as assessed using the repeated measures general linear model (GLM) with Sidak correction for multiple comparisons (Table 3). Figure 2, Figure 3 and Figure 4 illustrate these improvements graphically, emphasizing the statistical differences between the two groups over time.

A multivariate logistic regression analysis was conducted to identify factors associated with a three-point reduction in Visual Analog Scale (VAS) scores. The model’s fit was evaluated using the Hosmer and Lemeshow goodness-of-fit test (Table 4). The analysis revealed that lower body mass index (BMI) and lower Kellgren–Lawrence grade were significant predictors of achieving this outcome in both the C-PRP and A-PRP groups.

No serious adverse events were reported in either treatment group. Four patients before starting treatment, two in each group, decided to change therapy. The difference was not statistically significant (*p* = 0.76). The most common adverse event was acute, painful synovitis, which occurred in two patients in the C-PRP group and three patients in the A-PRP group, with symptoms resolving within an average of four days. Additionally, two patients in the C-PRP group experienced localized edema and difficulty walking, both of which resolved completely within seven days. None of the patients required hospitalization or arthrocentesis. Seventeen patients used acetaminophen for pain management, reporting either partial or complete relief. There were no statistically significant differences in the incidence of adverse events or treatment failure rates between the two groups (*p* = 0.68). A table with the main logistical differences of the two products is shown (Table 5).

## 4. Discussion

Knee osteoarthritis (OA) is a chronic degenerative joint disorder characterized by progressive cartilage degradation, synovial inflammation, and subchondral bone remodeling, leading to pain, stiffness, and functional impairment [27]. Current therapeutic strategies primarily focus on symptom management, but there is growing interest in biologic treatments, such as platelet-rich plasma (PRP), which may offer disease-modifying potential.

The findings of this study demonstrate that both umbilical cord-derived PRP (C-PRP) and autologous PRP (A-PRP) significantly reduced pain in patients with mild-to-moderate knee OA over a 12-month follow-up period. Notably, C-PRP exhibited superior pain relief in the short term (3–6 months), with a statistically significant difference observed at the 3-month mark. However, this advantage diminished over time, and by 9 and 12 months, pain scores were comparable between the two groups. The reduction in pain expressed according to the VAS score is also associated with a clinical-functional improvement. In fact, as demonstrated in the time course represented in Figure 1, Figure 2 and Figure 3, a significant improvement in the WOMAC and KOOS scores was highlighted in the first 6 months from the beginning of treatment, with higher values for the C-PRP group compared to the A-PRP. As for pain, functionality also showed a stationarity of the clinical state at 9–12 months with similar values between the two groups. These findings align with previous research suggesting that C-PRP may offer early symptomatic improvement but does not sustain superior pain relief compared to A-PRP in the long term.

Supporting this observation, Mazzotta et al. (2022) reported that C-PRP injections in hip osteoarthritis resulted in significant pain reduction within the first 2–6 months but failed to demonstrate clear superiority over A-PRP beyond this period [16]. Similarly, Caiaffa et al. (2021) found that C-PRP provided short-term clinical benefits in knee OA, though their study lacked long-term comparative data, leaving the sustained efficacy of C-PRP uncertain [27]. These findings suggest that while C-PRP may offer initial advantages in pain relief, it does not provide lasting benefits compared to A-PRP.

The short-term superiority of C-PRP may be attributed to its higher concentration of growth factors and regenerative components, which are thought to enhance tissue repair and modulate inflammation [28]. By delivering a more potent dose of these factors to the affected joint, C-PRP may induce a more pronounced initial therapeutic effect. This is consistent with the concept that the composition of PRP, including leukocyte concentration, plays a critical role in influencing clinical outcomes [12]. However, this early advantage of C-PRP was not sustained, as no significant differences were observed between the two groups at the 12-month assessment.

The convergence of outcomes at 12 months raises several important considerations. One possible explanation is that the sustained release of growth factors from A-PRP, despite its lower initial concentration, may eventually reach levels comparable to the initial bolus delivered by C-PRP. Alternatively, the progressive nature of osteoarthritis may limit the long-term efficacy of a single PRP injection, regardless of the source [13]. While PRP provides clinically significant benefits, its effects may wane over time, potentially necessitating repeat injections to maintain symptom relief and functional improvement [15]. This highlights the need for further investigation into optimal injection protocols and long-term efficacy.

Furthermore, Alessio-Mazzola et al. (2021) emphasized that the clinical response to PRP treatments is highly variable and influenced by factors such as patient BMI, OA severity, and platelet concentration [29], also evidenced by our logistic regression. Given the similar long-term pain scores observed in our study, it appears that C-PRP does not offer a clinically meaningful improvement in pain relief compared to the standard A-PRP protocol. This suggests that the hypothesized higher growth factor content of C-PRP may not translate into prolonged analgesic effects, or that A-PRP already provides a sufficient concentration of regenerative factors to achieve comparable outcomes over time [30].

In terms of functional improvement, both C-PRP and A-PRP led to significant enhancements in knee function over the 12-month follow-up period. However, similar to the pain relief outcomes, the initial advantage of C-PRP in functional improvement at the 3- and 6-month follow-ups diminished by the 9- and 12-month assessments. This finding is consistent with recent literature, which has failed to demonstrate a clear functional advantage of C-PRP over A-PRP in the long term. For example, Coskun et al. (2022) compared PRP with autologous conditioned serum (ACS) in knee OA and found that while biologic injections improved function, no single preparation consistently outperformed another in the long term [8]. We hypothesize that this lack of sustained differentiation may be related to patient-specific factors, such as baseline BMI or the severity of osteoarthritis, as previously suggested by Alessio-Mazzola et al. (2021) [29].

Both C-PRP and A-PRP were well-tolerated, with no serious adverse events reported in either group. The most common adverse event was acute, painful synovitis, which resolved within a few days. This is consistent with the safety profile reported in prior studies. For instance, Khurana et al. (2020) found that PRP injections were associated with minimal adverse effects, primarily localized pain and swelling, which resolved without intervention [31]. The absence of significant differences in adverse events between the two groups further supports the safety of both C-PRP and A-PRP as viable treatment options for knee OA [27].

In the context of orthopedic treatments, the cost of preparing Platelet-Rich Plasma (PRP) is a crucial factor influencing the choice of therapy, especially when comparing A-PRP and C-PRP. These two types of PRP differ significantly in their preparation processes, which consequently impacts their associated costs.

The preparation of A-PRP involves the collection of blood from the patient, followed by centrifugation to concentrate the platelets and growth factors. This process is relatively straightforward and can be performed with in-house equipment at a lower cost. In our study the cost of preparation for each A-PRP unit was 164 euros. The lower cost is influenced by the fact that the treatment is often carried out in local clinics or outpatient settings, with minimal logistical complexities.

In contrast, the preparation of C-PRP involves more intricate procedures. C-PRP is sourced from umbilical cord blood, requiring specialized facilities for collection, screening, and processing. The preparation of C-PRP is subject to stringent regulatory requirements, including ethical approvals, donor screening, and compliance with national and international standards for handling human tissues. These added layers of complexity necessitate the use of advanced laboratory facilities, resulting in a higher cost of preparation; The need for cryopreservation and additional quality control measures further contribute to the increased financial burden associated with C-PRP preparation. In our study the cost of preparation for each C-PRP unit was approximately 20% higher than the cost of A-PRP, amounting to 196 euros.

The aforementioned costs may vary significantly depending on the geographical area and on existing agreements between production centers and healthcare providers. However, the common higher cost of C-PRP preparation can be justified by its potential advantages in terms of the concentration of growth factors and anti-inflammatory cytokines, which may offer superior therapeutic outcomes in certain orthopedic conditions. However, despite these potential benefits, the increased cost of C-PRP remains a significant consideration for both healthcare providers and patients. While A-PRP offers a more accessible and cost-effective option, its effectiveness may be limited by patient-specific factors, such as platelet quality and the severity of the condition being treated. Conversely, C-PRP, although more expensive, may offer a more standardized and potentially more potent biological profile, which could translate into improved clinical outcomes in selected patient populations.

While this study provides understanding into the comparative efficacy of C-PRP and A-PRP, several limitations must be acknowledged. First, the sample size, though sufficient to detect overall improvements, may have limited our ability to identify subtle long-term differences between the two PRP types. Second, the absence of a placebo control group makes it challenging to definitively attribute the observed improvements solely to the PRP injections. As highlighted by Filardo et al. (2011), the placebo effect can play a significant role in studies involving injections, particularly in subjective outcomes such as pain and function [32].

The non-blinded nature of patient treatment selection may have introduced performance and expectation biases, potentially influencing subjective outcome measures such as pain perception and self-reported function. Patients aware of receiving a specific type of PRP (autologous vs. cord blood-derived) might have had preconceived notions about its efficacy, which could affect their reporting. Although objective measures were included, the lack of blinding remains a limitation and should be considered when interpreting the results.

## 5. Conclusions

The results of this study highlight the lack of strong evidence supporting C-PRP as a superior alternative to A-PRP in knee OA treatment. While C-PRP may provide initial pain relief, it does not lead to better long-term pain reduction or functional improvement. Given the higher costs, limited availability, and logistical challenges associated with umbilical cord-derived products, these findings suggest that A-PRP should remain the preferred PRP treatment modality for knee OA until more robust comparative studies are conducted.

## Figures and Tables

**Figure 1 jfmk-10-00138-f001:**
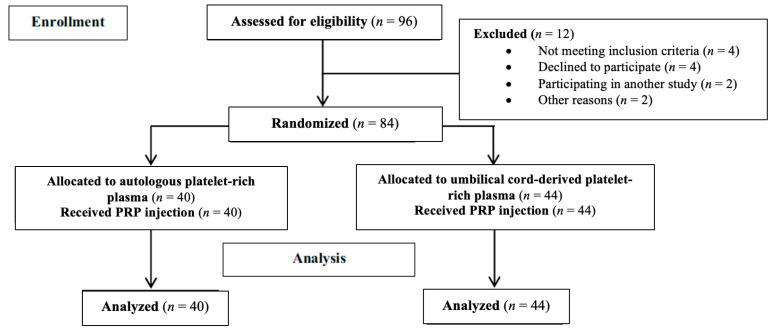
Diagram of the number of patients enrolled and analyzed in this study using STROBE guidelines.

**Figure 2 jfmk-10-00138-f002:**
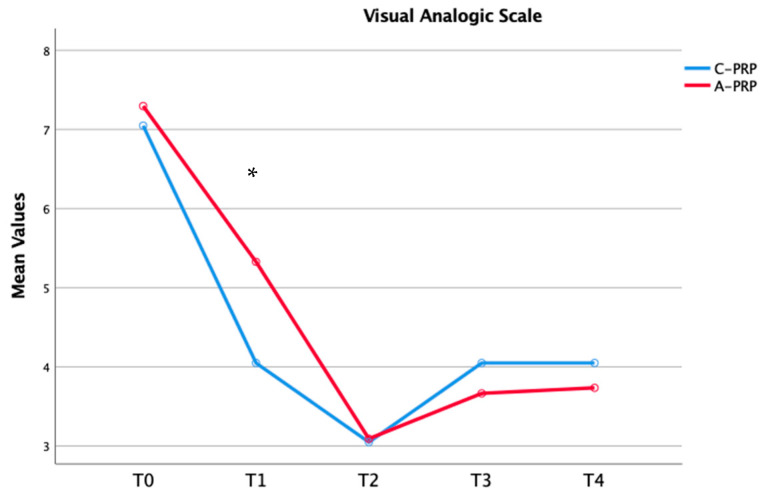
Temporal changes in median VAS scores across groups (* *p* = 0.01).

**Figure 3 jfmk-10-00138-f003:**
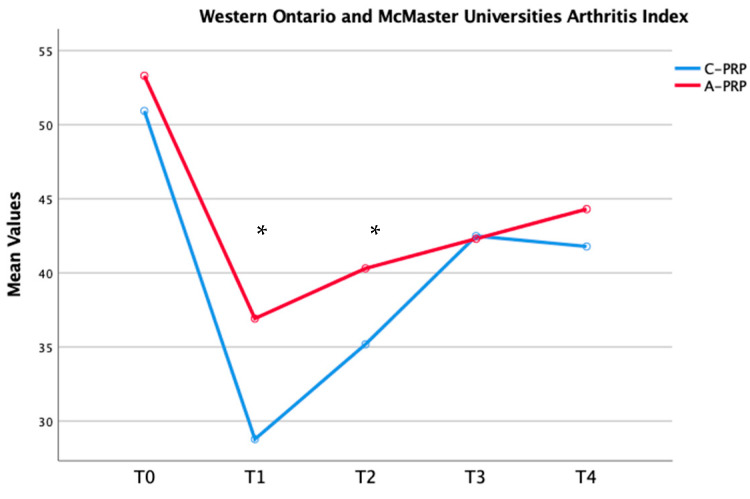
Temporal changes in median Western Ontario and McMaster University Osteoarthritis across groups (* *p* = 0.01).

**Figure 4 jfmk-10-00138-f004:**
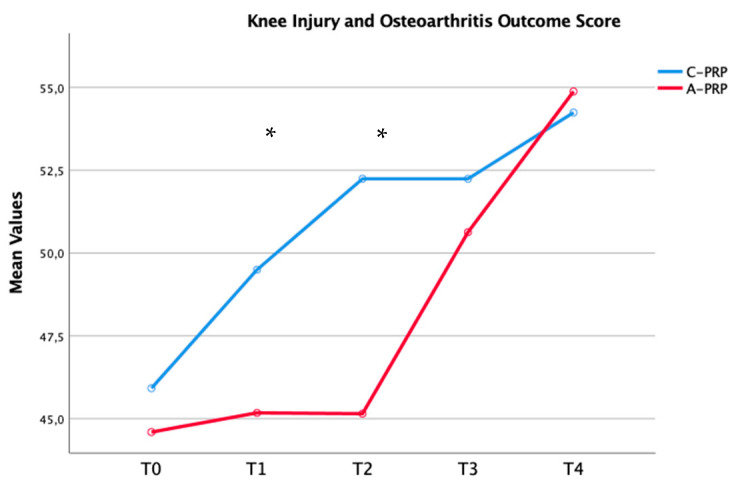
Temporal changes in median Knee Injury and Osteoarthritis Outcome Score across groups (* *p* = 0.01).

**Table 1 jfmk-10-00138-t001:** Main data of the study.

Preoperative Features	C-PRP Group (n = 44)	A-PRP Group (n = 40)	*p*-Value
**Age (year)**	58.36 ± 8.51	60.19 ± 9.21	0.23
**Gender (female)**	20 (45.5%)	22 (55%)	0.38
**BMI (Kg/m^2^)**	25.19 ± 5.27	25.36 ± 3.92	0.59
**Side (left)**	19 (43.2%)	15 (37.5%)	0.86
**Kellgren–Lawrence**			0.71
I	9 (20.5%)	10 (25%)	
II	20 (45.5%)	17 (42.5%)	
III	15 (34.1%)	13 (32.5%)	
**VAS_T0**	7.05 ± 1.31	7.29 ± 1.91	0.43
**WOMAC_T0**	50.93 ± 9.49	53.30 ± 13.43	0.39
**KOOS_T0**	45.92 ± 6.02	44.59 ± 4.15	0.75
**IKDC_T0**	40.24 ± 6.59	39.44 ± 4.53	0.61

Eighty-four patients; U Mann–Whitney and Fischer’s test; data are presented as mean ± standard deviation or number and percentage; BMI: Body Mass Index; VAS: Visual Analog Scale; WOMAC: Western Ontario and McMaster University Osteoarthritis Index; KOOS: Knee Injury and Osteoarthritis Outcome Score; IKDC: International Knee Documentation Committee Subjective Knee Form.

**Table 2 jfmk-10-00138-t002:** Differences in outcomes between groups.

		C-PRP Group (n = 44)	A-PRP Group (n = 40)	*p*-Value
**VAS**				
	T1	4.05 ± 1.23	5.23 ± 2.22	0.03
	T2	3.05 ± 0.94	3.09 ± 0.98	0.85
	T3	4.05 ± 0.89	3.66 ± 1.30	0.29
	T4	4.01 ± 0.68	3.73 ± 0.96	0.35
**WOMAC**				
	T1	28.78 ± 4.85	36.92 ± 10.15	0.01
	T2	35.18 ± 4.47	40.30 ± 13.43	0.03
	T3	42.48 ± 11.31	42.30 ± 12.42	0.94
	T4	41.78 ± 5.16	44.30 ± 13.35	0.29
**KOOS**				
	T1	49.50 ± 6.15	45.17 ± 4.41	0.01
	T2	52.24 ± 6.58	45.15 ± 6.71	0.01
	T3	52.12 ± 5.29	50.63 ± 7.21	0.20
	T4	54.24 ± 6.34	54.88 ± 6.70	0.54
**IKDC**				
	T1	53.24 ± 6.29	49.40 ± 5.48	0.07
	T2	53.67 ± 4.22	50.15 ± 6.71	0.01
	T3	48.78 ± 5.13	47.15 ± 6.74	0.70
	T4	47.25 ± 5.43	47.56 ± 8.31	0.88

Eighty-four patients; U Mann–Whitney test; data are presented as mean ± standard deviation; VAS: Visual Analog Scale; WOMAC: Western Ontario and McMaster University Osteoarthritis Index; KOOS: Knee Injury and Osteoarthritis Outcome Score; IKDC: International Knee Documentation Committee Subjective Knee Form.

**Table 3 jfmk-10-00138-t003:** Overall measure outcomes and variations between the initial and final f-u.

	Preoperative	Last Follow Up	Mean Difference	*p*-Value
**C-PRP Group** **(n = 44)**				
VAS	7.05 ± 1.31	4.01 ± 0.68	−3.04 ± 0.63	0.01
WOMAC	50.93 ± 9.49	41.78 ± 5.16	−9.23 ± 4.33	0.01
KOOS	45.92 ± 6.02	54.24 ± 6.34	8.32 ± 0.32	0.01
IKDC	40.24 ± 6.59	47.25 ± 5.43	7.01 ± 1.16	0.01
**A-PRP Group** **(n = 40)**				
VAS	7.29 ± 1.91	3.73 ± 0.96	−3.56 ± 0.95	0.01
WOMAC	53.30 ± 13.43	44.30 ± 13.35	−11.00 ± 0.08	0.01
KOOS	44.59 ± 4.15	54.24 ± 6.34	9.65 ± 2.19	0.01
IKDC	39.44 ± 4.53	47.56 ± 8.31	8.12 ± 3.78	0.01

Eighty-four patients; repeated measures general linear model with Sidak test; data are presented as mean ± standard deviation; VAS: Visual Analog Scale; WOMAC: Western Ontario and McMaster University Osteoarthritis Index; KOOS: Knee Injury and Osteoarthritis Outcome Score.

**Table 4 jfmk-10-00138-t004:** Multiple logistic regression model of factors influencing VAS reduction.

	OR	95% CI	SE	*p*-Value
**C-PRP Group** **(n = 44)**				
BMI (Kg/m^2^)	2.99	0.42–21.41	0.3	0.01
Kellgren–Lawrence	64.52	11.27–369.35	1.4	0.01
**A-PRP Group** **(n = 40)**				
BMI (Kg/m^2^)	3.56	0.98–32.29	0.6	0.01
Kellgren–Lawrence	59.46	10.13–289.78	1.2	0.01

**Table 5 jfmk-10-00138-t005:** Main logistical differences of the two PRP products.

Aspect	Autologous PRP (A-PRP)	Cord Blood-Derived PRP (C-PRP)
**Cost of Preparation**	€164 per session	€196 per session
**Availability**	Widely available in most orthopedic clinics and hospitals	Limited availability; requires specialized centers or biobanks
**Processing Complexity**	Simple; involves blood collection and centrifugation	Complex; involves umbilical cord blood collection, screening, and advanced processing
**Logistical Issues**	Minimal; processed at point of care, no need for external sourcing	High; requires specialized facilities, donor screening, cryopreservation, and transport
**Regulatory Considerations**	Minimal; standard blood processing regulations	Stringent; requires compliance with ethical, regulatory, and safety standards for human tissues
**Standardization**	Dependent on patient’s platelet count and blood quality	More standardized, with consistent quality due to controlled sourcing and processing
**Time to Preparation**	Relatively quick, usually within the same treatment session	Longer; involves additional steps such as donor screening, processing, and potential cryopreservation

## Data Availability

Data are unavailable due to privacy or ethical restrictions.

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
