# Peer review of "No Long-Term Superiority of Cord-Derived PRP over Autologous PRP in Knee Osteoarthritis: A Prospective Cohort Study"

_jfmk, 2025, doi:10.3390/jfmk10020138_

Round 1
Reviewer 1 Report
Comments and Suggestions for Authors
In this clinical trial, the authors found compared the clinical outcome of osteoarthritis patients receiving C-PRP (n-44) or A-PRP (n=40), and found that the C-PRP treatment had better clinical outcomes three months and six months after treatment, but similar at nine months and twelve months. The authors concluded that the A-PRP should remain the preferred PRP treatment for knee OA due to cost, availability and logistical challenge.
This clinical study is helpful for future research. However, addressing several questions would help improve this manuscript.
- In the introduction, I suggest the authors provide a more detailed summary of what’s currently known about the use C-PRP in the clinic, including results of recent clinical trial findings.
- In Table 3, the authors performed longitudinal comparison between the clinical outcome measures of T0 and T12, showing that at the final follow up, patient receiving both the C-PRP or A-PRP still has significant improvement before treatment. What is the rationale for the 12 month follow-up period? The authors should discuss whether this effect is consistent with reports that the PRP treatment typically stays effect for six to nine months.
- Would the author expect either C-PRP or A-PRP has a more long-lasting effect after 12-month based on current literature?
- A minor comment on Table 3: Why were there only three instead of four outcome measurement in the longitudinal comparison?
- The authors concluded that A-PRP should still be the preferred PRP treatment due to cost, availability and logistical challenge. I would recommend the authors to include a table comparing C-PRP and A-PRP regarding these aspects to illustrate this argument.
Author Response
This clinical study is helpful for future research. However, addressing several questions would help improve this manuscript.
- In the introduction, I suggest the authors provide a more detailed summary of what’s currently known about the use C-PRP in the clinic, including results of recent clinical trial findings. The introduction has been completely revised by including this suggestion.
- In Table 3, the authors performed longitudinal comparison between the clinical outcome measures of T0 and T12, showing that at the final follow up, patient receiving both the C-PRP or A-PRP still has significant improvement before treatment. What is the rationale for the 12 month follow-up period? The authors should discuss whether this effect is consistent with reports that the PRP treatment typically stays effect for six to nine months. The introduction has been completely revised by including this suggestion.
- Would the author expect either C-PRP or A-PRP has a more long-lasting effect after 12-month based on current literature? The introduction has been completely revised by including this suggestion including new references.
- A minor comment on Table 3: Why were there only three instead of four outcome measurement in the longitudinal comparison? The Table 3 has been revised by including this suggestion.
- The authors concluded that A-PRP should still be the preferred PRP treatment due to cost, availability and logistical challenge. I would recommend the authors to include a table comparing C-PRP and A-PRP regarding these aspects to illustrate this argument. The Table 5 has been included in the paper.
Reviewer 2 Report
Comments and Suggestions for Authors
The manuscript entitled "No Long-Term Superiority of Cord-Derived PRP Over Autologous PRP in Knee Osteoarthritis: A Prospective Cohort Study" has been reviewed. This manuscript addresses a clinically relevant topic. Several areas require improvement to enhance the rigor and impact of the manuscript.
Patients were allowed to switch treatment groups. This compromises the internal validity and introduces allocation bias. The authors should clarify how many participants switched groups and whether an intention-to-treat analysis was considered.
Discussion: more emphasis should be placed on what constitutes a minimal clinically important difference in VAS, WOMAC, and KOOS scores, to interpret the real-world significance of findings.
Additional discussion about potential biases introduced by the non-blinded nature of patient treatment selection is warranted.
Comments on the Quality of English LanguageThe English could be improved to more clearly express the research.
Author Response
The manuscript entitled "No Long-Term Superiority of Cord-Derived PRP Over Autologous PRP in Knee Osteoarthritis: A Prospective Cohort Study" has been reviewed. This manuscript addresses a clinically relevant topic. Several areas require improvement to enhance the rigor and impact of the manuscript.
- Patients were allowed to switch treatment groups. This compromises the internal validity and introduces allocation bias. The authors should clarify how many participants switched groups and whether an intention-to-treat analysis was considered.
Patients who did not agree with the randomized choice of PRP could change before treatment and at the end of the study it was not possible to change therapy in case of failure. We added and analyzed this information in the results and discussed in the body of the manuscript. We would specify that a total of four patients, two from each group requested to change treatment before the study
- Discussion: more emphasis should be placed on what constitutes a minimal clinically important difference in VAS, WOMAC, and KOOS scores, to interpret the real-world significance of findings.
- Additional discussion about potential biases introduced by the non-blinded nature of patient treatment selection is warranted.
Added and explained in the end of discussion.
Round 2
Reviewer 2 Report
Comments and Suggestions for Authors
Thank you for responding to the comments. The updates were appropriate and I have nothing further to suggest.